# Adaptive immune defense prevents *Bartonella* persistence upon trans-placental transmission

**Lena K. Siewert**[1,2¤], **Christoph Dehio**[1☉]*, **Daniel D. Pinschewer**[2☉]*

**1** Biozentrum, University of Basel, Basel, Switzerland, **2** Division of Experimental Virology, Department of Biomedicine, University of Basel, Basel, Switzerland

☉ These authors contributed equally to this work.
¤ Current address: Experimental Neuroimmunology, Department of Biomedicine, University Hospital of Basel, Basel, Switzerland
* christoph.dehio@unibas.ch (CD); daniel.pinschewer@unibas.ch (DDP)

**Data Availability Statement:** All relevant data are within the manuscript and its Supporting Information files.

**Funding:** C.D. was funded by grant 310030B_201273 by the Swiss National Science

## Abstract

Vertical transmission of *Bartonella* infection has been reported for several mammalian species including mice and humans. Accordingly, it is commonly held that acquired immunological tolerance contributes critically to the high prevalence of Bartonellae in wild-ranging rodent populations. Here we studied an experimental model of *Bartonella* infection in mice to assess the impact of maternal and newborn immune defense on vertical transmission and bacterial persistence in the offspring, respectively. Congenital infection was frequently observed in B cell-deficient mothers but not in immunocompetent dams, which correlated with a rapid onset of an antibacterial antibody response in infected WT animals. Intriguingly, B cell-deficient offspring with congenital infection exhibited long-term bacteremia whereas B cell-sufficient offspring cleared bacteremia within a few weeks after birth. Clearance of congenital *Bartonella* infection resulted in immunity against bacterial rechallenge, with the animals mounting *Bartonella*-neutralizing antibody responses of normal magnitude. These observations reveal a key role for humoral immune defense by the mother and offspring in preventing and eliminating vertical transmission. Moreover, congenital *Bartonella* infection does not induce humoral immune tolerance but results in anti-bacterial immunity, questioning the contribution of neonatal tolerance to *Bartonella* prevalence in wild-ranging rodents.

## Author summary

Vertical transmission of *Bartonella* has been reported in small rodents but also in at least one human case. The prevalence of these bacteria in the wild is extremely high. While a protective antibody response clearly controls the infection in the experimental model, observations from the wild indicate that this might not always be the case. This led to the long-standing hypothesis that *Bartonella* might induce immunological tolerance. To study if transplacental transmission of these bacteria results in immunological tolerance in the offspring, we used a mouse model of *Bartonella taylorii* infection. We infected

Foundation; https://www.snf.ch/en. D.D.P was funded by Hans Buss Stiftung, Basel-Stadt, Switzerland. https://www.fundraiso.ch/en/sponsor/hans-buss-stiftung. The funders had no role in study design, data collection and analysis, decision to publish, or preparation of the manuscript.

**Competing interests:** The authors of this manuscript have the following competing interests: Daniel D. Pinschewer is a founder, shareholder and consultant of Hookipa Pharma Inc.

wildtype and immunocompromised females (Rag1-/- and μMT) and observed that transmission only occurred in mothers lacking a functional B-cell response. Immunocompetent offspring, however, cleared the infection and were protected from reinfection by the same *Bartonella* strain due to the presence of protective antibodies. Thus, even though transplacental transmission of *Bartonella* is possible under the right circumstances, we find no evidence for immunological tolerance or persistent infection in the offspring.

## Introduction

Bartonellae are gram-negative, facultative intracellular bacteria, which are transmitted by blood-sucking arthropods [1–3], establish long-lasting intra-erythrocytic bacteremia and can cause significant human disease [4–6]. Antibody responses to *Bartonella* have been reported from infected cats [7], mice [8,9] and humans [10–14], and we have recently shown that the clearance of *Bartonella* bacteremia in mice is accompanied by a neutralizing antibody response preventing bacterial attachment to red blood cells [9]. In contrast, antibody measurements in *Bartonella*-infected wild-captured rodents detected only low or undetectable genogroup-specific responses [15–17], whereas the prevalence of active *Bartonella* infection in wild-caught rodents can reach 65% [18]. These and related findings [15] led to the hypothesis that infected animals in the wild differ from experimentally infected animals with respect to their *Bartonella*-specific immune responsiveness. Systems serology studies in *Bartonella*-infected cats demonstrated, however, that only about 7% of the bacterial proteome is targeted by the host antibody response [19], revealing important limitations of earlier immunoreactivity studies utilizing whole bacterial lysates as antigenic assay substrate [15–17]. These few antigenic targets in natural infection were disproportionately localized in the bacterial membrane, which is important in light of recent discoveries on the hypervariability of a key antibody target domain in a *Bartonella* outer membrane protein [9]. In conjunction with frequent gene transfers by the gene transfer agent (GTA) [20,21] as a source of genetic variability of *Bartonella* in the wild, these new insights question the utility of genogroup-specific whole bacterial cell lysates for an assessment of *Bartonella* seroprevalence [15,16].

Specific immunological tolerance in the offspring owing to vertical transmission was proposed as a mechanism of supposedly life-long persistent *Bartonella* infection [15]. Observations made in the 1930s on immunological tolerance to a virus transmitted *in utero* [22] have contributed to the formulation of the clonal selection theory, and the experimental validation of the concept by *in utero* administered allografts culminated in the 1960 Nobel Prize to Burnet and Medawar for the concept of immunological tolerance [23,24]. In light of this it was intriguing that *Bartonella* spp. could be isolated from embryos and neonates born to captured rodents that carried the bacteria but lacked specific antibodies [15]. Transplacental transmission itself was experimentally validated in a murine model of *Bartonella* infection, yet was observed to cause fetal loss and resorption, supposedly as a result of the maternal immune response [25]. Analogously to mice, vertical transmission of *Bartonella* has been reported in a human patient [26] but has not been observed in infected cats [27] or cattle [28].

Here we report that upon vertical transmission only immunocompromised offspring developed persistent bacteremia whereas immunocompetent offspring clear the infection and develop protective immune memory. These data argue against neonatal immunological tolerance as a main mechanism of *Bartonella* persistence and widespread infection in the wild.

## Results

### Efficient transplacental *B. taylorii* transmission in B cell-deficient but not wildtype mice

To investigate transplacental transmission of *Bartonella* and its outcome in the offspring, we exploited the *B. taylorii* infection model of C57BL/6 mice. It offers the advantage that further to quantification of bacteremia in blood a bacterial neutralization assay has been developed allowing for the quantification of protective antibodies in the sera of infected mice [9]. To address the possibility that maternal immune defense interfered with bacterial transmission we compared WT mice with gene knock-out mouse strains defective in B cells (μMT) or lacking both T and B cells (Rag1-/-) (Fig 1A). We infected dams with $10^7$ cfu *i.d.* and mated them ten days later with uninfected males, an experimental setup that has previously been used to study vertical transmission in a murine model of *Bartonella* infection [25]. This timing of infection relative to mating warranted that the entire gestation occurred in the bacteremic

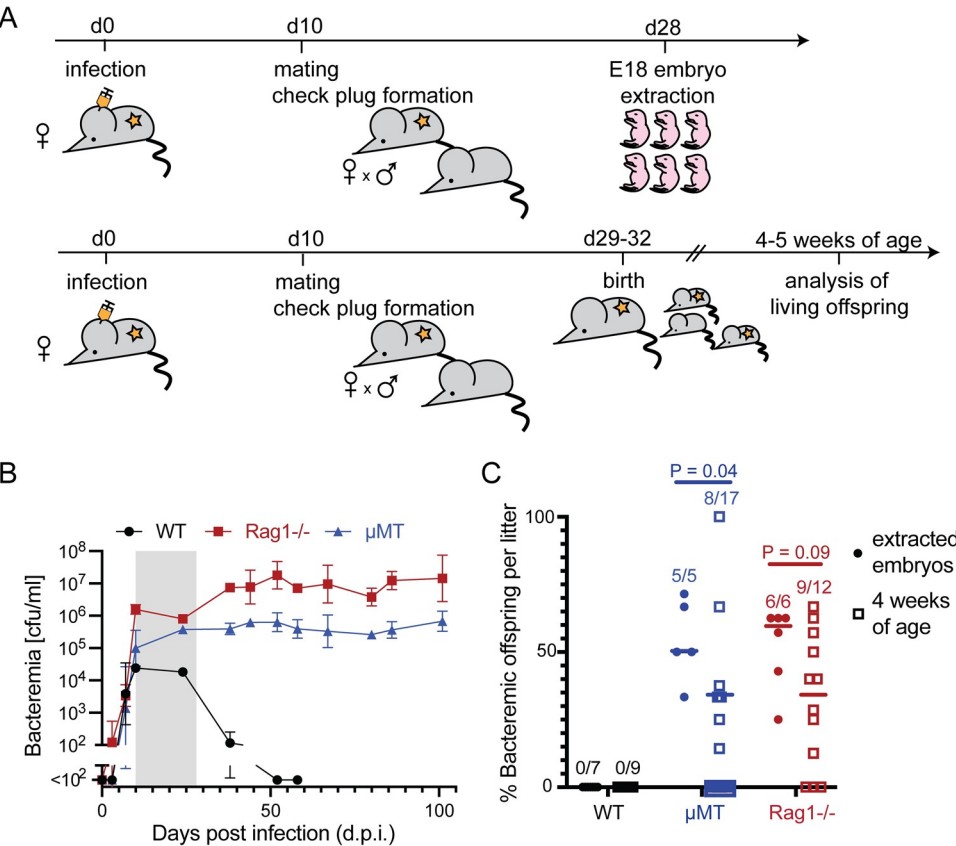

**Fig 1. Efficient transplacental *B. taylorii* transmission in B cell-deficient but not wildtype mice. (A)** Schematic overview of the experimental setup for the analysis of extracted embryos (top) and living offspring (bottom). Female WT, Rag1-/- and μMT mice were infected with $10^7$ cfu of *B. taylorii* i.d. and were mated 10 days later. Embryos were extracted on day 18 of gestation and offspring were evaluated at 5 weeks of age. **(B)** Bacteremia of WT, Rag1-/- and μMT mice after inoculation with $10^7$ cfu *B. taylorii* i.d.. Symbols represent the mean ± SD of three mice per group. One representative of three experiments is shown. The time window for pregnancy of WT mice (upon mating on day 10 post infection) is indicated in grey. **(C)** The percentage of WT, Rag1-/- and μMT offspring harbouring *B. taylorii* was determined at embryonic day 18 ("extracted embryos") and at four to five weeks after birth ("4–5 weeks of age"). Numbers above symbols in **(C)** indicate the percentage of litters containing at least one infected embryo or litter mate, respectively/number of litters assessed. Each symbol represents one litter, horizontal lines indicate the mean. Unpaired two-tailed Student's t-tests were performed for statistical analysis. Related data are reported in S1 Fig.

**Table 1. Transplacental transmission of *B. taylorii* in B cell-deficient, but not wildtype mice.**

| Genotype of the mother[1] | Embryo extraction[2] | | | | |
|---|---|---|---|---|---|
| | # of analyzed litters | % of culture-positive offspring per litter; mean ± SD (range) | # of analyzed embryos | # of culture-positive embryos | # of culture-positive placentae |
| WT | 7 | 0 ± 0 (0–0) | 48 | 0 | 18 |
| μMT | 5 | 50.0 ± 15.2 (33.3–71.4) | 31 | 17 | 25 |
| Rag1-/- | 6 | 59.8 ± 15.3 (25.0–62.5) | 42 | 22 | 41 |
| | Offspring 4–5 weeks of age[3] | | | | |
| | # of analyzed litters | % of bacteremic offspring per litter; mean ± SD (range) | # of analyzed pups | # of bacteremic pups | % bacteremic pups |
| WT | 9 | 0 ± 0 (0–0) | 47 | 0 | 0 |
| μMT | 17 | 33.3 ± 28.6 (0–100) | 54 | 15 | 27.8 |
| Rag1-/- | 12 | 34.3 ± 24.7 (0–66.7) | 68 | 12 | 17.6 |

[1] Dams were infected with $10^7$ cfu *B. taylorii* i.d. and were mated 10 days later with uninfected males. For the analysis of 4-5-week-old Rag1-/- and μMT offspring, persistently infected Rag1-/- and μMT dams were also mated continuously.

[2] Total numbers for extracted embryos and placentae were determined on day 18 of gestation.

[3] Living offspring were analyzed at 4-5 weeks of age.

period of WT animals (Fig 1A and 1B). As expected, WT but not B cell deficient dams mounted antibacterially protective erythrocyte adhesion inhibiting (EAI) antibodies [9] in the course of their pregnancy (S1A and S1B Fig). When analyzing embryos on day 18 of gestation (day 28 of the experiment), we failed to detect Bartonellae in any one of the 48 embryos extracted from a total of 7 WT litters, indicating vertical transmission of *B. taylorii* was highly inefficient in WT animals in this setting and did not lead to a productive transmission of bacteria (Fig 1C and Table 1). In contrast, *Bartonella* infection was detected in at least one embryo from 6 out of 6 Rag1-/- litters analyzed, with a total of 22 out of 42 embryos culture-positive (Table 1). Similarly, 5 out of 5 μMT litters analyzed by embryo extraction contained at least one infected embryo, with positivity rates within litters ranging from 33.3–71.4%. These differences in vertical transmission rates between WT and immunocompromised mothers suggested that the maternal and/or fetal immune systems can interfere with *Bartonella* transmission.

Next we analyzed the living offspring at 4–5 weeks of age (Fig 1C). The percentage of bacteremic μMT offspring per litter was significantly lower than determined for extracted embryos, and an analogous statistical trend was noted for Rag1-/- offspring (Fig 1C). While an average of 50.0% μMT embryos and 59.8% Rag1-/- embryos were culture-positive, only 33.3% of μMT pups and 34.3% of Rag1-/- pups were infected at 4–5 weeks of age, albeit with considerable variation between litters (Fig 1C and Table 1). We noted that litters of infected Rag1-/- dams were significantly larger at the time point of embryo extraction than observed on the day after birth and a similar trend, although not statistically significant (p = 0.1068), was noted for litters of μMT mothers but not for WT controls (S1C and S1D and S1E Fig), which correlated with bacterial transmission in Rag1-/- and μMT, but not WT mice.

## Lack of correlation between bacterial burden in maternal blood and embryos

Next we aimed to determine if the bacterial burden in maternal blood and/or the placenta was decisive for vertical *Bartonella* transmission. In keeping with the kinetics of bacteremia in WT, μMT and Rag1-/- mice (Fig 1B) [8,9], bacterial loads in the blood of μMT females at the time point of mating were similar to WT mice, whereas bacteremia in Rag1-/- females was higher

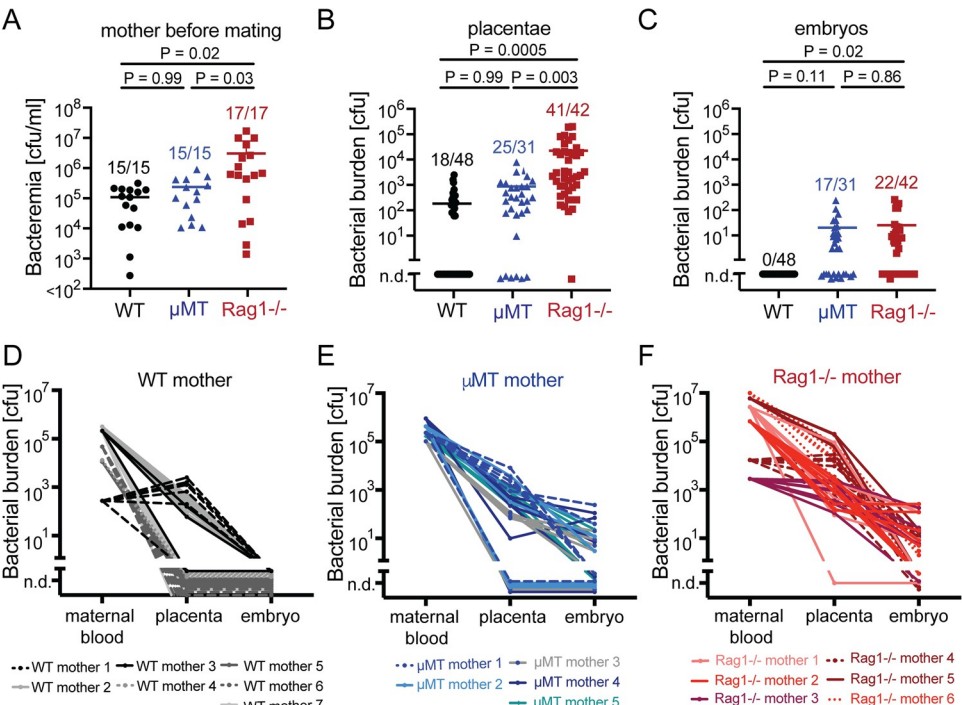

**Fig 2. Lack of correlation between bacterial burden in maternal blood, placentae and embryos.** Female WT mice were infected with $10^7$ cfu *B. taylorii* i.d. and mated with WT partners on day 10 as reported in Fig 1 (the same animals as in Fig 1C are shown). Embryos and placentae were extracted on day 18 of gestation. Female μMT and Rag1-/- mice were infected with *B. taylorii* $10^7$ cfu i.d. and mated continuously from day 10 onwards with partners of the same genotype. **(A)** Bacteremia of the mothers at the time point of mating (WT mice) or first mating (μMT and Rag1-/- mice) was determined. **(B-C)** The amount of cultivatable bacteria from placentae **(B)** and embryos **(C)** was also measured. Numbers above symbols in **(A-C)** indicate the number of bacteria-containing specimens/number of specimens tested. **(D-F)** The relationship between maternal bacterial blood titer, the bacterial burden in placentae and embryos is depicted for WT (D), μMT (E) and Rag1-/- (F) mothers. All symbols and lines represent individual mice, placentae and embryos, horizontal lines in **(A-C)** show the mean. Between-group differences were analyzed by two-way ANOVA with Tukey's multiple comparisons test. Resulting *P*-values are indicated. Related data are reported in S1 Fig.

than in the other two groups (Fig 2A). An assessment of bacterial burden in the placenta on day 18 of gestation followed an analogous pattern (Fig 2B and Table 1): 18 out of 48 WT placentae (36.7%) contained detectable bacteria at levels that were in the range of those 25 out of 31 μMT placentae (80.6%) that harbored *Bartonella*. Bacteria were also detected in all but one Rag1-/- placenta analyzed (41/42, 97.7%), with on average higher bacterial loads than in the placentae of WT and μMT dams (Fig 2B). It seems possible if not likely, however, that this placental bacterial load consisted at least partially in residual maternal blood. While WT embryos remained culture-negative, the bacterial burden of infected Rag1-/- and μMT embryos was in similar ranges despite somewhat differential levels of maternal bacteremia and placental bacterial loads (Fig 2C, compare Fig 2A and 2B). Accordingly, a comparison of maternal bacteremia with placental and embryonic bacterial burden at the level of individual dams and embryos did not evidence a clear quantitative correlation of these three parameters (Fig 2D and 2E), except that all infected embryos had an infected placenta, too. Taken together, these data suggested that maternal bacteremia determined the average bacterial burden in the placenta, whereas maternal antibody responses prevented *Bartonella* infection of the embryo irrespective of the bacterial load observed in the placenta.

## Immunocompetent offspring clear vertically transmitted *Bartonella* infection and develop protective immunity

Next, we aimed to study the immunological outcome in congenitally infected offspring. We mated persistently infected Rag1-/- and μMT females with partners that were either WT or of the respective same genotype. The resulting offspring born to these dams were, therefore, either immunocompetent Rag1+/- and μMT+/- mice or immunocompromised Rag1-/- and μMT-/- animals, respectively (Fig 3A). Analogously to the adult infection of Rag1-/- and μMT-/- mice, vertical transmission to animals of the same genotype resulted in life-long bacteremia (Fig 3B and 3C; compare Fig 1B) [8,9]. Intriguingly, however, immunocompetent heterozygous Rag1+/- and μMT+/- offspring cleared *Bartonella* infection within 5 to 11 weeks after birth (Fig 3B and 3C). This observation suggested that the offspring's adaptive immune defense prevented the life-long persistence of vertically transmitted *B. taylorii* infection in a B cell-dependent manner. With the aim of identifying immunological correlates of anti-*Bartonella* immunity in vertically transmitted infection we evaluated antibacterial antibody responses of immunocompetent μMT+/- offspring that had been born to bacteremic μMT mothers. We hypothesized that bacteremic offspring mounted protective anti-bacterial B cell responses to control *Bartonella* infection. To test this, we employed our recently described erythrocyte adhesion inhibition (EAI) assay, which can be used to quantitatively assess protective *Bartonella*-specific antibody responses in the serum of mice [9]. Low-level serum EAI titers were detected in four out of 21 abacteremic μMT+/- offspring and in none of the 14 littermates exhibiting detectable bacteremia at 4–5 weeks of age (Fig 3D). These findings raised the possibility that vertically transmitted bacteria triggered an antibacterial B cell response that was either weak or not well-maintained, or that our assay system was insufficiently sensitive to consistently detect these responses. Hence, we tested whether congenitally infected offspring, after clearance of *B. taylorii*, were immune to re-challenge with the pathogen (Fig 3E). Mice that had cleared vertically transmitted infection remained sterile upon challenge whereas all littermates with undetectable bacterial loads after birth developed bacteremia (Fig 3F). One and two weeks post challenge we collected serum to determine EAI titers. At one week after challenge six out of 15 μMT+/- mice with congenital infection and three out of 11 littermates without congenital infection mounted detectable EAI responses. Within two weeks after *Bartonella* challenge, functional anti-*Bartonella* antibody responses were detectable in almost all μMT+/- animals, irrespective of postnatal bacteremia (11/11 congenitally infected, 14/15 litter mates; Fig 3G). Taken together these results showed that mice with vertically transmitted *B. taylorii* infection cleared the bacteria. Moreover, they formed protective immunological memory, which prevented *Bartonella* bacteremia upon re-challenge. Unimpaired anti-*Bartonella* antibody responses to bacterial re-exposure indicated further, that animals with congenitally acquired *Bartonella* infection had not developed durable immunological tolerance to the pathogen.

## Discussion

This study deciphers the role of the adaptive immune defence in curtailing vertical *B. taylorii* transmission and bacterial persistence in the infected offspring. Not only does the maternal B cell response restrict transmission but the immunocompetent offspring's immune defence also clears the infection, protects against reinfection and mounts a functional antibody response upon bacterial re-challenge. It appears, therefore, that counter to earlier assumptions there is no discernible humoral immune tolerance to *Bartonella* upon *in utero* transmission.

The concept that antigen exposure *in utero* or perinatally results in immunological tolerance [23,24] is substantially influenced by early observations from mice congenitally infected

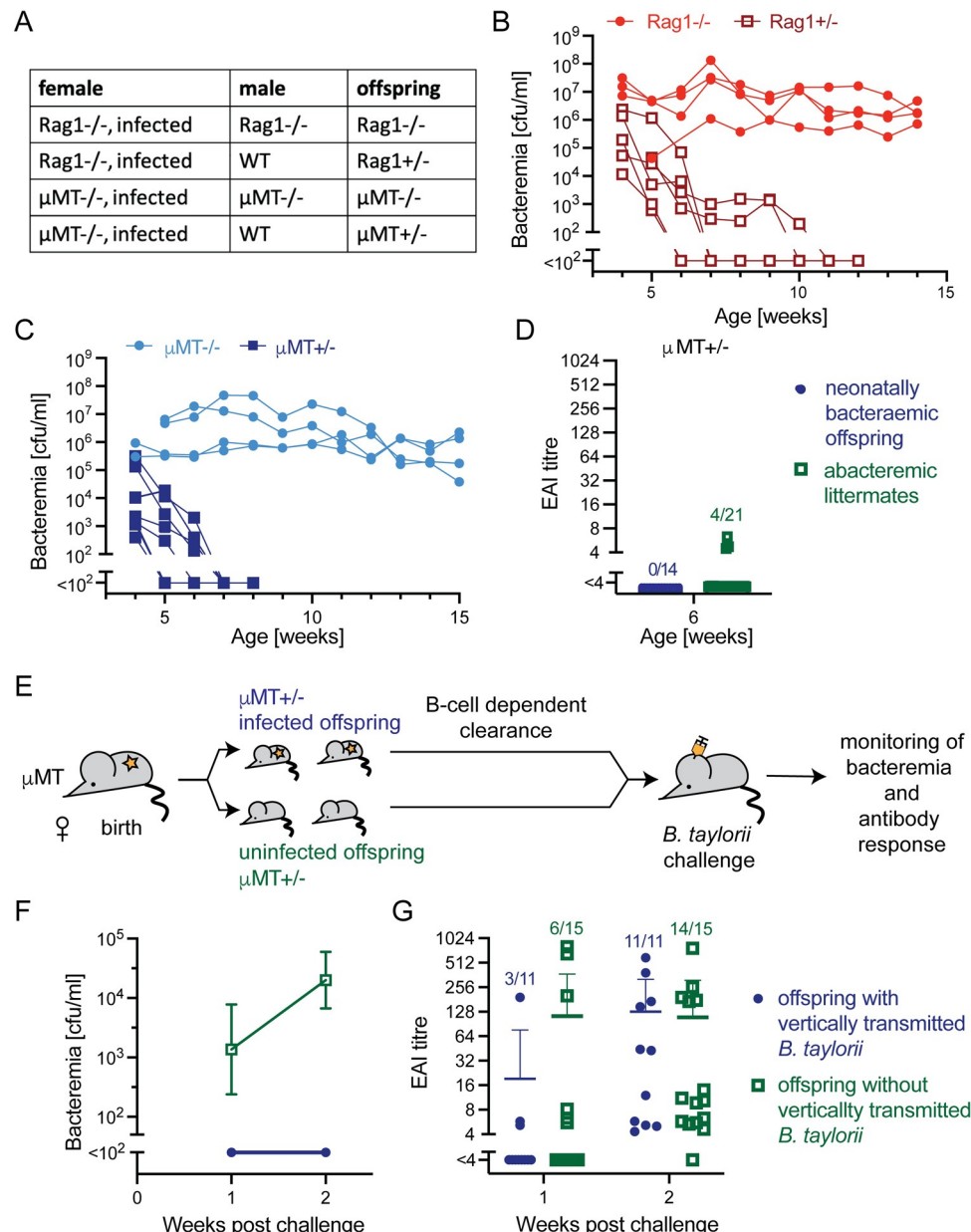

**Fig 3. Immunocompetent offspring clear vertically transmitted *Bartonella* infection and develop protective immunity to re-infection. (A)** Schematic overview of homozygous (immunocompromized) and heterozygous (immunocompetent) crosses between infected Rag1-/- or μMT dams and uninfected males of the indicated genotypes. Female mice were infected i.d. with $10^7$ cfu of *B. taylorii* and were mated continuously with either WT, Rag1-/- or μMT-/- males. **(B-C)** Bacteremia of immunocompetent (μMT+/-, Rag1+/-) and immunocompromised (μMT-/-, Rag1-/-) pups born to Rag1-/- (B) and μMT (C) mothers. **(D)** The antibody response of immunocompetent μMT +/- offspring at 6 weeks of age was determined by means of an erythrocyte adhesion inhibition (EAI) assay [9], comparing animals with vertically transmitted *B. taylorii* infection and uninfected litter mates. **(E)** Schematic overview for the challenge experiment presented in (F-G): Offspring infected with *B. taylorii* by vertical transmission were re-challenged with with $10^7$ cfu of the same bacterial strain after they had cleared the congenital infection. Litter mates without detectable vertical *B. taylorii* infection served as controls. Bacteremia **(F)** and EAI antibody titers **(G)** were determined at the indicated time points. The data represents pooled results from: (B) 10 animals (4 Rag1-/- and 6 Rag +/-), (C) 13 animals (4 μMT -/- and 9 μMT +/-), (F-G) 26 animals (11 with congenital infection, 15 without congenital infection). Numbers above symbols in **(D, G)** indicate the number of animals with EAI antibody titers above technical cut-off/number of animals tested. Symbols in **(B, C, D, G)** show individual mice, data points in **(F)** represent the mean ± SD of 14 and 8 animals, respectively.

with lymphocytic choriomeningitis virus (LCMV) [22] and later has found validation in perinatal hepatitis B virus (HBV) infection of humans [29]. Here we report that immunocompetent mice clear vertically acquired *Bartonella* infection in a B cell-dependent manner and develop protective immunity against bacterial re-challenge. Acquired immunological tolerance is best validated for the clonal elimination of CD8 T cell precursors in the thymus as elegantly demonstrated in LCMV carrier mice [30], whereas central CD4 T cell tolerance to the same virus seems incomplete [31,32]. Accordingly, neither murine carriers of LCMV nor persistently HBV-infected humans exhibit solid humoral immune tolerance to the viruses they carry, and antiviral antibody formation is commonly observed [33–39]. From a purely immunological standpoint the lack of humoral immune tolerance to congenital *Bartonella* infection may, therefore, largely resemble the situation in congenital LCMV and HBV infections. The main difference may rather consist in the predominantly humoral immune-mediated control of *Bartonella* infection [8,9], whereas LCMV and HBV can only be eliminated when a potent effector CD8+ T cell response is mounted [40,41]. Perinatally acquired CD8 T cell tolerance may thus result in LCMV and HBV persistence whereas *Bartonella* is effectively eliminated by the incompletely tolerized humoral arm of the immune system.

The failure to detect antibody responses in wild-captured *Bartonella*-infected rodents [15,16] and the presence of *Bartonella* in their embryos and neonates [15] have led to speculations that these bacteria exploit transplacental transmission to cause life-long tolerant infection. As discussed above, our data argue against neonatal tolerance as a mechanism that substantially contributes to the high prevalence of Bartonellosis in wild-ranging rodents. Instead, our earlier reports have documented that *Bartonella* modulates the host's innate immune response to its benefit [42] and that antigenic variation of key antibody targets is presumed to facilitate repeated infections of the same rodent host by closely related bacterial strains [9], altogether warranting the evolutionary success of these bacteria in the wild.

Congenital *Bartonella* infection has been observed in embryos and neonatal mice born to naturally infected, wild-captured white-footed mice and cotton rats [15]. In contrast, an earlier report on a murine model of transplacental *Bartonella birtlesii* transmission in experimentally infected BALB/c mice described fetal resorptions and a reduced body weight of viable fetuses, which was supposedly due to *Bartonella*-induced placental vasculitis and resulting placental insufficiency [25]. Unlike for wild-captured white-footed mice [15], however, vertical *B. birtlesii* transmission in experimentally infected BALB/c mice was limited to fetal resorptions, without any evidence for bacterial transmission to living offspring. In keeping with the latter findings, our study on *Bartonella taylorii* infection in C57BL/6 mice suggests vertical transmission is restricted to B cell-deficient dams. In contrast to the *B. birtlesii* infection model in BALB/c mice, however, we did not find any embryos undergoing resorption when dissecting the uteri for extraction of embryos and placentae. While differences in the occurrence and possibly the timing of prenatal lethality may be due to a differential inflammatory response profile of the inbred mouse strains tested [43], we propose that maternal antibodies could interfere with bacteria crossing the placental barrier, similar to the prevention of red blood cell adhesion [9]. Hence, the kinetics of the antibacterial antibody response may contribute to the probability of vertical transmission and its outcome.

We observed that Rag1-/- litters on the day after birth were smaller than at the time point of embryo extraction (E18), and a similar trend, although not statistically significant, was noted for μMT-/- litters but not for WT litters. This finding may indicate that congenital *Bartonella* infection may negatively impact the offspring's perinatal fitness. The reduced prevalence of *Bartonella* bacteremia in 4-5-week-old μMT mice as compared to extracted μMT embryos and an analogous trend in Rag1-/- mice provide independent support for this hypothesis.

Limitations of our study consist in the largely undefined role of maternal and fetal T cells in vertical *Bartonella* transmission. Higher bacterial loads in Rag1-/- as compared to μMT-/- mothers likely explain a trend towards higher transmission rates in the former and suggest the role of T cells in *Bartonella* control extends beyond the mere help to B cells. The contribution of individual immunoglobulin isotypes to the prevention of vertical transmission and to the control of bacteremia in the offspring, respectively, have not been addressed by our study either. While the IgM response of activation-induced deaminase- (AID-) deficient mice has been shown to effectively control *B. taylorii* in adult mice without a need for isotype class switch [9], the ability of IgG to cross the placenta may improve its antibacterial efficacy at the maternal-fetal interface. Last but not least the lack of transplacental *Bartonella* transmission by WT mothers in our model precluded us from analyzing a potential role of breast milk antibodies in preventing bacteremia in the offspring.

Taken together the present findings demonstrate that B cells represent a dual layer of immune protection against vertical transmission of *B. taylorii*, and possibly of other *Bartonella* species, in mice. Maternal B cell responses generally prevent infection of the offspring *in utero* and, if accidentally overrun, the B cell system of the newborn represents a fail-safe mechanism for the prevention of bacterial persistence during an otherwise immunologically vulnerable period of life.

## Materials and methods

### Ethics statement

Animal handling was performed in accordance with the Swiss Animal Protection law and under the auspice of the institutional animal welfare officers. All animal experiments were performed at the University of Basel, with permission by the Veterinary Office of the Canton Basel-Stadt (license 1741_29584).

### Cultivation of bacteria

*Bartonella taylorii* IBS 296 Sm$^R$ WT (LSB001) [44] and GFP+ (LSB115) [9] were grown at 35°C, 5% $CO_2$ on Columbia blood agar (CBA; Oxoid Cat#CM0331) containing 5% defibrinated sheep blood (Oxoid, Cat#SR0051), 100 μg/ml streptomycin (AppliChem, Cat#A2951) and 10 μg/ml gentamycin (AppliChem, Cat#A11492) if required. Bacteria were streaked as thumbnails for three days followed by a subsequent expansion on a fresh plate for 2 days prior to usage. The *B. taylorii* IBS 296 strain was used for this study since it readily establishes robust bacteremia in mice and its immunobiology in mice has been thoroughly characterized [9,45]. Moreover, a GFP-expressing variant is available, thereby enabling the functional EIA antibody readout reported in Figs 3G and S1A and S1B. Streptomycin resistance (Sm$^R$) encoded by a mutation in *rpsL* facilitates targeted genetic manipulation [46], but is not expected to impact the bacterium's behavior or immunobiology in mice, neither does this strain feature any other genetic alterations that are known or expected to bias the results reported in this study.

### Animal experimentation

5 week old WT C57Bl/6JRj and Balb/cJRj mice were obtained from Janvier labs, France. Rag1-/- [47] were bred at the Laboratory Animal Science Center (LASC, University of Zurich, Switzerland). μMT mice [48] were obtained from Jackson Laboratories, USA, and a colony was maintained locally at the University of Basel, Switzerland.

Mice were anesthetized with isoflurane to administer $10^7$ cfu *B. taylorii i.d.* into the ear. At each time point, for which bacteremia is reported in the figures, 50 μl of blood was collected

from the tail vein in a sterile manner into 3.8% Na-citrate (Merck, Cat#71497). An interval of at least one week was respected between blood draws, except for the first two weeks after adult infection (reported in Figs 1B and S1A and S1B), during which two weekly blood samples were collected. Blood was frozen at -80˚C. After thawing serial dilutions were plated of CBA blood plates containing streptomycin and incubated at 35˚C 5% $CO_2$ for 5–7 days.

For embryo extraction, females were mated on day 10 post infection and were monitored daily for plug formation. Embryos and placentae were extracted in a sterile manner on day 18 of gestation. After a collagenase digest (30min at 37˚C in 3 mg/ml Collagenase IV (Gibco, Cat#17104019), 2% FCS (Amimed, Cat#2-01F30I) in DMEM (Gibco, Cat#61965), the digested embryos and placentae were pushed through a cell strainer into DMEM containing 10% FCS. Samples were frozen at -80˚C for erythrocyte lysis and bacterial release, and after thawing were plated in dilution series on CBA blood plates containing streptomycin and incubated at 35˚C, 5% $CO_2$ for 5–7 days.

For the analysis of living offspring, Rag1-/- and μMT females were mated continuously (from day 10 post infection onwards) with either WT, Rag1-/- or μMT males respectively. The offspring were weaned at 3 weeks of age and blood was collected starting at 4–5 weeks of age.

## Erythrocyte adhesion inhibition (EAI) assay

The EAI assay was performed as described in detail elsewhere [9]. In brief, erythrocytes obtained from Balb/cJRj mice were purified using a Ficoll gradient (Ficoll-Paque, VWR, Cat#17-1440-02) and were stored for up to 2 weeks in DMEM (Gibco, Cat#61965) containing 10% FCS FCS (Amimed, Cat#2-01F30I) at 4˚C prior to use.

Serial dilutions of sera were performed in 96-well U-bottom plates (Greiner, Cat#650161) in DMEM containing 10% FCS. $5x10^5$ cfu *B. taylorii* expressing GFP were added per well and the plates were incubated at 35˚C, 5% $CO_2$ for 1h prior to the addition of $10^6$ red blood cells (multiplicity of infection, MOI = 0.5) in 100 μl DMEM containing 10% FCS. The next day, the supernatant was removed, the red blood cells were fixed using 1% PFA (EMS, Cat#EMS-15710) and 0.2% gluturaldehyde (EMS, Cat#16020) in PBS (BioConcept, Cat#3-05F29-I) for 10 min at 4˚C in the dark. After quenching with 2% FCS in PBS, the cells were analysed for GFP signal by Flow Cytometry (BD Canto II using HTS autosampler).

## Data analysis

Statistical analysis of the obtained data was performed using GraphPad Prism Software. Student's t test was used for pair-wise comparisons, one-way ANOVA with Bonferroni's post-test was performed for the comparison of more than two groups. $P < 0.05$ was considered statistically significant, $P < 0.01$ as highly significant, $P \leq 0.1$ was interpreted as a statistical trend.

## Supporting information

**S1 Fig. The antibody response against *B. taylorii* in adult animals and the size of litters in utero and born to infected dams. (A-B)** We infected female WT (A) and μMT (B) mice and determined bacteremia and erythrocyte adhesion inhibiting (EAI) antibody titers at the indicated time points. The mice and bacteremia curves are the same ones as displayed in Fig 1B. Symbols represent the mean ± SEM of three mice per group. One representative of three experiments is shown. **(C-E)** We infected WT **(C)**, μMT **(D)** and Rag1-/- **(E)** dams with $10^7$ cfu of *B. taylorii* i.d. and mated them with partners of the same genotype 10 days later (analogously to the experiment in Fig 1). The size of litters was recorded at the time point of embryo extraction on embryonic day 18 ("extracted") or within 24 h after birth ("after birth"). Symbols in **(A-B)** show mean ± SD of combined data from two independent experiments with at least

three mice per group. (C-E) Each symbol represents an individual litter, horizontal lines and error bars depict the mean ± SD. Statistical analysis was performed by unpaired Student's t-test, with P-values indicated in the figure.
(TIF)

**S1 Table. Raw data table.** Excel spreadsheet containing, in separate sheets, the underlying numerical data and statistical analysis for Fig panels 1B, 1C, 2B-F, 3B-D, 3F and 4G and S1A-E.
(XLSX)

## Author Contributions

**Conceptualization:** Lena K. Siewert, Christoph Dehio, Daniel D. Pinschewer.

**Funding acquisition:** Christoph Dehio, Daniel D. Pinschewer.

**Investigation:** Lena K. Siewert.

**Methodology:** Lena K. Siewert.

**Project administration:** Lena K. Siewert.

**Writing – original draft:** Lena K. Siewert, Daniel D. Pinschewer.

**Writing – review & editing:** Lena K. Siewert, Christoph Dehio, Daniel D. Pinschewer.

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
