## [Decision Letter · Decision Letter 0]

2 Mar 2022

Dear Prof. Pinschewer,

Thank you very much for submitting your manuscript "Adaptive immune defense prevents Bartonella persistence upon trans-placental transmission" for consideration at PLOS Pathogens. As with all papers reviewed by the journal, your manuscript was reviewed by members of the editorial board and by several independent reviewers. In light of the reviews (below this email), we would like to invite the resubmission of a significantly-revised version that takes into account the reviewers' comments.

While two of the reviewers request only minor clarifications, reviewer 3 has requested further dissection of the immune cells/antibody types contributing to the phenotype. While the experiments in points 1, 2, and 4 would increase impact of the work, they could be addressed in the discussion as limitations of the current work. Point 3 regarding the role of milk IgA in elimination of blood infection in WT offspring, however, should be clarified and can be done with existing reagents.

We cannot make any decision about publication until we have seen the revised manuscript and your response to the reviewers' comments. Your revised manuscript is also likely to be sent to reviewers for further evaluation.

Sincerely,

Congli Yuan

Guest Editor

PLOS Pathogens

Nina Salama

Section Editor

PLOS Pathogens

Kasturi Haldar

Editor-in-Chief

PLOS Pathogens

orcid.org/0000-0001-5065-158X

Michael Malim

Editor-in-Chief

PLOS Pathogens

orcid.org/0000-0002-7699-2064

Reviewer's Responses to Questions

**Part I - Summary**

Reviewer #1: This study explores the role of humoral immune tolerance versus anti-bacterial immunity in vertical/maternal transmission and persistence of Bartonella in mice. The findings of this study reveal a key role for humoral immune defense by the mother and offspring in preventing and eliminating vertical transmission of Bartonella taylorii in mice. These are interesting and novel contributions to the knowledge of the immune response to Bartonella infections.

The questions explored here are well studied and documented. They are of interest to the Bartonella and bacteriology research community, and within the scope of PLOS Pathogens.

Reviewer #2: State of the art story excluding vertical transmission of B. taylorii in mice.

Reviewer #3: In this work, Siewert et al have investigated the frequency of vertical transmission of Bartonella as a function of the ability to generate Bartonella-neutralizing antibodies of both mothers and their offspring. B-cells and antibodies are shown to be required to limit bacteremia and placental bacterial burden in the mothers on the one hand, and is additionally required to successfully fend off infection in the offspring. The data refute a previous model whereby persistent Bartonella infection was due to immune tolerance to the infection of the offspring. The data are strong and the experiments well controlled.

**Part II – Major Issues: Key Experiments Required for Acceptance**

Reviewer #1: None

Reviewer #2: - none -

Reviewer #3: While concise, the data as currently presented could be improved by the addition of a bit more detail on the following aspects:

1. What is the role of T-cells? The consistent difference in bacterial burden and transmission efficacy between Rag1-/- and uMT mice suggests that T-cells have a role as well. What is the phenotype of T-cell deficient mice in this system?

2. Which antibody class is most important in humoral defense? Please use ko's specifically lacking IgG or IgA to address this question.

3. How important are milk antibodies in preventing bacteremia in infected offspring? Swapping of litters from infected to naive dams, depriving the offspring of potentially protective IgA in milk, would address this question.

4. Finally, how effective are antibodies in WT offspring of heterozygous uMT parents at successfully fending off transmission?

**Part III – Minor Issues: Editorial and Data Presentation Modifications**

Reviewer #1: L 120 – the word "mating" is more appropriate than "impregnation".

L 153 – "bacterial loads" should be used instead "bacterial titers".

L 246 – should be suggests.

L – 256 – I strongly suggest the authors to add the words "although not significant" after the phrase "and a similar trend". In addition, in line 258 the authors should change to "….and possibly also in µMT-/- mice but not in WT….", as P=0.7209 is not considered significant (although stated by the authors as a trend) – otherwise it might be considered as overestimation of the test results.

L 262-3 – Authors conclude :" Taken together the present findings demonstrate that B cells represent a dual layer of immune protection against vertical transmission of Bartonellae". Authors should be more accurate regarding their findings as their findings are restricted to B. taylorii in mice. I therefore strongly recommend changing the sentence to the following :" Taken together, the present findings demonstrate that B cells represent a dual layer of immune protection against vertical transmission of B. taylorii, and possibly of other Bartonella species, in mice".

L 286 – authors indicate that "blood was collected from the tail vein" – authors should include the time points, frequency of blood collection and how much blood was drawn from each mouse in each time-point. This information is important as it may affect the general health/stress condition of the mice.

L205-6 – the authors wrote :"Not only does the maternal B cell response restrict transmission." This is a fragment that cannot stand by its own. It should be merged with the next sentence in the paragraph.

L 290 – delete the word days.

L 401 – delete the word as.

Reviewer #2: 1. Introduction is too long - please shorten to ~ 66%.

2. line 268-274: please explain: why was a smR mutant used instead of wildtype bacteria? How was this mutant characterized? Are there any other mutations in this particular strain which might artifically influence the experiments?

3. bacterial load was identified by culture from -80°C frozen samples. Although I know that something like this has been published before, I am critical about this procedure. Did the authors compare these data to data obtaind from fresh samples which underwent direct cultivation?

4. Suppl, Fig. 1: Fig. A and B: please give group sizes. Please wrtite out "EAI" as this abbreviation is difficult to find in the text.

Reviewer #3: none

PLOS authors have the option to publish the peer review history of their article (what does this mean?). If published, this will include your full peer review and any attached files.

Reviewer #1: No

Reviewer #2: No

Reviewer #3: No
---

## [Editor Report · Decision Letter 1]

1 Apr 2022

Dear Prof. Pinschewer,

We are pleased to inform you that your manuscript 'Adaptive immune defense prevents Bartonella persistence upon trans-placental transmission' has been provisionally accepted for publication in PLOS Pathogens.

Best regards,

Congli Yuan

Guest Editor

PLOS Pathogens

Nina Salama

Section Editor

PLOS Pathogens

Kasturi Haldar

Editor-in-Chief

PLOS Pathogens

orcid.org/0000-0001-5065-158X

Michael Malim

Editor-in-Chief

PLOS Pathogens

orcid.org/0000-0002-7699-2064
---

## [Editor Report · Acceptance letter]

19 Apr 2022

Dear Prof. Pinschewer,

We are delighted to inform you that your manuscript, "Adaptive immune defense prevents Bartonella persistence upon trans-placental transmission," has been formally accepted for publication in PLOS Pathogens.

Best regards,

Kasturi Haldar

Editor-in-Chief

PLOS Pathogens

orcid.org/0000-0001-5065-158X

Michael Malim

Editor-in-Chief

PLOS Pathogens

orcid.org/0000-0002-7699-2064